# First-Trimester Biochemical Serum Markers in Female Kidney Transplant Recipients—The Impact of Graft Function

**DOI:** 10.3390/ijerph192316352

**Published:** 2022-12-06

**Authors:** Natalia Mazanowska, Patrycja Jarmużek-Orska, Bronisława Pietrzak, Joanna Pazik, Zoulikha Jabiry-Zieniewicz, Przemysław Kosiński

**Affiliations:** 1First Department of Obstetrics and Gynecology, Medical University of Warsaw, pl. Starynkiewicza 1/3, 02-015 Warszawa, Poland; 2Department of Transplantation Medicine and Nephrology, Medical University of Warsaw, 02-015 Warszawa, Poland; 3Department of Obstetrics, Perinatology and Gynecology, Medical University of Warsaw, 02-091 Warszawa, Poland

**Keywords:** prenatal screening, posttransplant pregnancy, kidney transplantation, first-trimester combined screening

## Abstract

Data on serum biochemistry markers as a component of the first-trimester screening test in pregnant kidney graft recipients are limited. In the absence of a separate validated algorithm, biochemical testing is commonly used in the first-trimester screening in kidney transplant recipients. Therefore, the study aimed to analyze first-trimester serum biochemical markers and the first trimester combined screening results in pregnant kidney graft recipients. A retrospective study was carried out in pregnant women who underwent the first-trimester combined screening test performed per the Fetal Medicine Foundation (FMF) protocol in 2009–2020. The study group included 27 pregnancies in kidney graft transplant recipients, and the control group was 110 patients with normal kidney function, matched according to age, body mass index (BMI), and gestational age. The biochemical serum markers (free beta-human chorionic gonadotropin [beta-hCG] and pregnancy-associated plasma protein A [PAPP-A]) were evaluated using the FMF-approved Roche Elecsys^®^ assay and exhibited as multiples of the median (MoM) values. Data on first-trimester screening test results, perinatal outcomes, and graft function (assessed using serum creatinine concentrations) were analyzed. The analysis of first-trimester screening parameters revealed no difference in nuchal translucency (NT) measurements and uterine artery flow. However, free beta-hCG MoM and PAPP-A values were higher in posttransplant pregnancies than in controls: 3.47 ± 2.08 vs. 1.38 ± 0.85 (*p* = 0.035) and 1.46 ± 0.81 vs. 0.98 ± 0.57 (*p* = 0.007), respectively. The false positive rate of trisomy 21 (T21) screening in graft recipients was 25.9% vs. 3% in the controls. The free β-hCG MoM values positively correlated with serum creatinine levels before (r = 0.653; *p* < 0.001), during (r = 0.619; *p* = 0.001), and after pregnancy (r = 0.697; *p* < 0.001). There was a statistically significant negative correlation for PAPP-A MoM values for postpartum serum creatinine concentration (r = −0.424, *p* = 0.035). Our results show significantly higher serum concentrations of free beta-hCG and PAPP-A in posttransplant pregnancies than in healthy controls, confirmed when exhibited as MoM values and their association with graft function was assessed by serum creatinine concentration. Taking those changes into account would reduce the high number of false positive test results in this group. The validated first-trimester screening algorithm that considers altered kidney function in pregnant kidney graft recipients remains to be developed.

## 1. Introduction

The first report of pregnancy after kidney transplantation was published in 1963, and since then, thousands of posttransplant pregnancies have been reported worldwide [1,2]. In many cases, the kidney transplant procedure in young end-stage kidney disease patients leads to fertility restoration and pregnancy. However, posttransplant pregnancy remains high-risk, and a successful outcome is possible only with a multidisciplinary approach and individualized care. The prospective data are limited, and the management is based mainly on knowledge derived from national transplant registries, single-center studies, and published case series. The risks, apart from exacerbation of prepregnancy diseases such as chronic hypertension and diabetes, include, but are not limited to, premature birth, intrauterine growth restriction, cesarean delivery, and development of hypertensive disorders of pregnancy, as well as graft failure [3,4].

The primary aim of non-invasive prenatal screening is to find a high-risk group for the most frequent chromosomal abnormalities (trisomy 21,18, and 13). The combined first-trimester screening is based on maternal age and history, ultrasound measurement of the thickness of fetal nuchal translucency (NT) performed between 11 + 0 and 13 + 6 weeks of gestation, and the maternal serum concentrations of free β-human chorionic gonadotrophin (free β-hCG) and pregnancy-associated plasma protein-A (PAPP-A). Ultrasound NT assessment allows the detection of approximately 75–80% of fetuses with an abnormal karyotype, with a false positive rate (FPR) of 5% [5]. The biochemical serum markers’ evaluation incorporated as an integral part of the test improves the performance of first-trimester screening by up to 90%, with an FPR of 3%. In all three trisomies, the PAPP-A concentration is decreased, while free β-hCG concentration is usually decreased in T18 and T13 and increased in T21 pregnancies [6]. The free β-hCG and PAPP-A concentrations vary with gestational age and are influenced by maternal characteristics such as weight, ethnicity, smoking status, method of conception, or diabetes mellitus [7]. Therefore, the measured concentrations are converted to gestational age-adjusted multiples of the expected median (MoM) values, validated for a specific analytical system.

Data on the serum biochemistry results in pregnant kidney graft recipients are limited. However, other serum markers widely used in clinical practice, such as serum tumor markers, increase in posttransplant patients, supposedly in relation to renal function impairment [8]. It is well-known that human choriogonadotropin (hCG) is cleared by kidneys and elevated in chronic kidney disease (CKD) patients [9]. Similarly, there are reports of elevated serum PAPP-A concentrations in the non-pregnant kidney recipient population (males and females) correlated with c-reactive protein elevation and impaired kidney function (serum creatinine, urea, and uric acid levels) [10].

Currently, there is still no consensus on the performance of first-trimester combined screening in posttransplant pregnancies. The individual risk of chromosomal aberrations is calculated according to the Fetal Medicine Foundation (FMF) algorithm for the general population without taking kidney function into account. In the absence of a separate validated algorithm, biochemical testing is commonly used in the first-trimester screening in kidney transplant recipients. Therefore, the study aimed to analyze first-trimester serum biochemical markers and the first-trimester combined screening results in pregnant kidney graft recipients.

## 2. Materials and Methods

A retrospective study was carried out in a group of pregnant women who underwent routine ultrasound scans at 11–13 + 6 weeks in the Ist Department of Obstetrics and Gynaecology of the Medical University of Warsaw and continued care in our department until delivery. From 2009 until 2020, we identified 27 pregnancies in kidney graft transplant recipients who underwent a combined first-trimester screening test. The control group included 110 healthy patients (with normal kidney function) matched according to age, BMI, and gestational age. All patients had the first-trimester combined screening test performed per the FMF protocol, beginning with a questionnaire on maternal age, racial origin, method of conception, smoking, parity, and medical history. Maternal weight and height were taken on the day of the scan. A blood sample for free beta-hCG and PAPP-A was taken. These biochemical markers were evaluated using a Roche Elecsys^®^ assay (Mannheim, Germany) approved by the FMF. All ultrasound scans were performed transabdominally with a Voluson E6 ultrasound machine (GE Healthcare, Vienna, Austria) operated by an experienced sonographer, holder of the Certificate of Competence, in the 11 + 0 to 13 + 6-weeks scan and Doppler scan issued by the Fetal Medicine Foundation (). Ultrasound markers such as NT and the uterine artery pulsatility index (UtA PI) were assessed per the FMF protocol. Patients were given their adjusted individual risk for trisomy 21, 18, and 13, and those with a high risk (>1:300 according to Polish Society of Obstetricians and Gynecologists guidelines) were given the option of invasive testing for fetal karyotype. Data on pregnancy outcomes were collected from the hospital maternity records or directly from patients. Graft function was assessed using serum creatinine concentrations for background prepregnancy (up to six months before conception), first trimester, and the first week postpartum.

Statistical analysis: Statistical comparison of variables between study and control groups was carried out. Data were presented as mean values. Pearson’s correlation was used to investigate the correlation of variable factors, and *p* < 0.05 was considered statistically significant. All statistical analyses were performed with IBM SPSS statistics software (SPSS Inc., Chicago, IL, USA).

## 3. Results

The characteristics of the study and control groups are presented in Table 1.

Both groups were similar in baseline maternal characteristics (age, BMI, and parity). The study group patients had a higher incidence of chronic hypertension, pregnancy complications, cesarean section, and preterm delivery. The average time from transplant surgery to delivery was 8.9 ± 4.5 years, and for three women, it was a repeated surgery. The immunosuppressive regimens were primarily tacrolimus-based (70.4%), with the remainder administered cyclosporin A and one patient azathioprine only. Kidney function assessed using serum creatinine concentration remained satisfactory throughout the pregnancy, with no acute graft rejection episodes. The median baseline and pregnancy laboratory parameters of the study group are presented in Table 2. The perinatal outcomes (Table 1) show a higher incidence of preterm delivery, with a lower mean birth weight, but without a statistically significant difference in small-for-gestation babies, defined as a birth weight below the 10th centile.

The analysis of the first-trimester screening parameters (Table 3) revealed no difference in nuchal translucency measurements and uterine artery flow. However, serum biochemical marker (PAPP-A and free beta-hCG) concentrations were higher in posttransplant pregnancies than in controls, in both absolute values and when expressed as multiples of the median (MoM).

The subgroup analysis, taking into account tacrolimus or cyclosporine-A-based regimens, revealed no statistically significant difference in first-trimester serum biochemical markers, as well as in perinatal outcomes or graft function (Table 4).

Tacrolimus-treated patients demonstrated, however, a statistically significant increase in serum creatinine concentration over time when prepregnancy values (1.04 ± 0.28 mg/dL) were compared to those during pregnancy (1.18 ± 0.39 mg/dL, *p* = 0.038) and postpartum (1.23 ± 0.44 mg/dL, *p* = 0.07), which was not the case in the CsA-treated subgroup (0.99 ± 0.11; 0.99 ± 0.32; 0.95± mg/dL, respectively, *p* = NS) (Figure 1).

We also analyzed serum association between serum creatinine concentration and biochemical serum markers expressed as MoM values in kidney graft recipients. The free β-hCG MoM values positively correlated with serum creatinine levels before (r = 0.653; *p* < 0.001), during (r = 0.619; *p* = 0.001), and after pregnancy (r = 0.697; *p* < 0.001). For PAPP-A MoM values, we found a statistically significant negative correlation only for postpartum serum creatinine concentration (r = −0.424, *p* = 0.035). In the study group, PAPP-A MoM values were also significantly higher in patients that did not develop pregnancy-induced hypertension later on (1.59 ± 0.86 vs. 1.01 ±0.40 mg/dL, *p* = 0.032).

The individual risk of trisomy was estimated as high (>1:300) in 7 out of 27 posttransplant pregnancies (with the highest risk > 1:4 in one patient), with no cases of abnormal fetal karyotype. This accounts for 25.9% of false positive test results versus 3% in the controls.

## 4. Discussion

Our study aimed to evaluate first-trimester biochemical serum markers in pregnant kidney graft recipients and the performance of first-trimester combined screening in this unique group of patients. Our results show significantly higher serum concentrations of free β-hCG and PAPP-A in posttransplant pregnancies than in healthy controls, confirmed when exhibited as MoM values. What is more, the false positive high-risk test results were found in 25.9% of pregnancies, whereas the population FPR threshold is set at 3–5%, depending on the screening protocol and how many ultrasound markers are considered [11].

The increase in β-hCG concentration has been described in the posttransplant population. Our results indicate that the median free β-hCG value in the study group was 3.47 ± 2.08 MoM, which is 2.5 times more than in healthy controls. In a series of eleven women with renal transplant, published in 2013, the median value for free β-hCG was 2.15 MoM [12]. The abnormal renal function resulting in the elevation of beta-hCG, used as a second-trimester biochemical screening marker for T21, was also described in two series of 12 and 14 kidney transplant pregnant women, with average beta-hCG levels of 3.0 and 2.73 MoM [13,14]. Kidneys excrete beta-hCG; therefore, detectable levels were reported in men and non-pregnant women with severe kidney disease [9]. The studies on supra elevated serum free β-hCG levels (reaching 5 MoMs) in low-risk pregnancies suggest their association with impaired maternal renal function, expressed as the lower estimated glomerular filtration rate (eGFR) or higher serum creatinine concentration [15,16]. Our results confirm the statistically significant correlation between first-trimester free β-hCG MoM values and serum creatinine concentrations measured before, during, and after pregnancy (r = 0.653, *p* < 0.001; r = 0.619, *p* = 0.001; r = 0.697, *p* < 0.001, respectively).

In trisomy 21, the average levels of free β-hCG between 11 and 13 + 6 weeks of gestation are close to 2.0 MoM, in contrast to 1.0 MoM in the case of normal fetal karyotype [6]. The elevation of free β-hCG might therefore lead to an increase in false positive screening test results. Our false positives rate of 25.9% was substantially higher than in the general population. Similarly, the previously cited studies reporting elevated free beta-hCG levels in posttransplant pregnancies observed the high prevalence of false positives, equal to 27% in the first-trimester series [12]. Another two studies on the T21 screening test employing beta-hCG concentration in the second trimester had rates of false positives of 66% and 14%, respectively [13,14].

Interestingly, the paper by Grande et al. investigates three models for adjusting the false positives rates [12]. It was proposed to adjust free b-hCG levels according to the deviation of serum creatinine concentration, employing three different methods (median, proportionality, and regression) described in detail in the paper. The non-adjusted 27% false positive rate dropped after the re-estimation of the T21 risk to 18% or even 10%, depending on the model used. The authors suggested that the regression method might be the most applicable. Although posttransplant pregnancy is regarded as high-risk, the risks do not include an increased prevalence of congenital anomalies or chromosomal aberrations. Our analysis reveales no differences in NT measurement compared to healthy controls (1.77 ± 0.43 vs. 1.70 ± 0.47, *p* = NS), in accordance with the results by Valentin et al. in a population of patients with renal disease [16]. Therefore, the risk elevation could be attributed to shifts in serum biochemical marker concentrations related to maternal kidney function. Grande et al. concluded that there is a need for a validated method for adjusting free beta-hCG levels during the first trimester according to serum creatinine [12]. It would enable us to lower the high false positive results and avoid invasive procedures in this group of patients.

In our cohort, PAPP-A MoM values were also significantly increased compared to healthy controls (1.46 ± 0.81 vs. 0.98 ± 0.57 MoM; *p*= 0.007). Similar results were published by Grande et al. in the earlier mentioned series of 11 kidney transplant pregnancies (1.41 vs. 1.07 MoM) [12]. In contrast to that study, where no significant correlation was found between PAPP-A and creatinine levels, we found a statistically significant negative correlation for postpartum serum creatinine concentration (r = −0.424, *p* = 0.035). The lower first-trimester PAPP-A values in our posttransplant cohort were associated with a higher postpartum serum creatinine concentration and more frequent pregnancy-induced hypertension development. Inadequate placentation, reflected by low first-trimester maternal serum PAPP-A, is a causal factor of hypertension development in pregnancy [17]. As the kidney graft recipient population has a high prevalence of hypertensive disorders in pregnancy, one might presumably expect lower PAPP-A values. On the other hand, PAPP-A is not only a valuable tool in the first-trimester combined screening but also serves as a marker of vascular damage. Its elevation is associated with impaired renal function and proteinuria and indicates adverse outcomes in hemodialysis and posttransplant patients [18]. High levels of PAPP-A are closely associated with high serum levels of urea, creatinine, uric acid, and inflammatory markers [10]. It is reasonable to assume that partially impaired renal function in pregnant kidney graft recipients and vascular damage could mask trophoblast invasion disorders in early pregnancy, thus falsely elevating decreased background PAPP-A levels. However, our cohort did not manifest one of the biophysical markers of impaired placenta perfusion resulting from inadequate trophoblast invasion—the increase in uterine artery resistance. There was no difference in the uterine artery pulsatility index in the study and control groups (1.06 ± 0.23 vs. 1.28 ± 0.53; *p* = NS).

Whether these phenomena influence the reliability of the combined test or early prediction of preeclampsia in kidney graft recipients similarly to elevated free beta-hCG levels remains to be established.

The perinatal outcomes in our posttransplant population confirm those by Transplant Pregnancy Registry International, with a trend towards more preterm deliveries and more babies delivered via cesarean section [2]. As confirmed by other published data, we found no differences in perinatal outcomes according to the type of calcineurin inhibitor administered to the kidney recipients [4].

The main limitation to our study is a retrospective analysis of posttransplant pregnancies that ended in delivery, with no data on early pregnancy loss. On the other hand, to our knowledge, this is the largest published cohort of kidney posttransplant pregnancies concentrating on the clinical utility of the first-trimester combined test that is routinely used in the general population.

## 5. Conclusions

To sum up the characteristics of serum biochemical markers after kidney transplantation, beta-hCG increases in the presence of impaired kidney function. At the same time, an increase in PAPP-A can be associated with both kidney graft dysfunction and inflammation/vascular damage and a decrease with impaired placentation, presumably more often appearing in kidney transplant pregnancies. Taking those changes into account would reduce the high number of false positive test results in this group. It is essential as other methods of non-invasive prenatal testing, such as cell-free fetal DNA tests, are also problematic in pregnant women with a kidney transplant. They demonstrate three types of circulating cell-free DNA: maternal, deriving from the trophoblast tissue, and kidney graft, which might influence fetal aneuploidy assessment [19,20]. The validated first-trimester screening algorithm that considers altered kidney function in pregnant kidney graft recipients remains to be developed.

## Figures and Tables

**Figure 1 ijerph-19-16352-f001:**
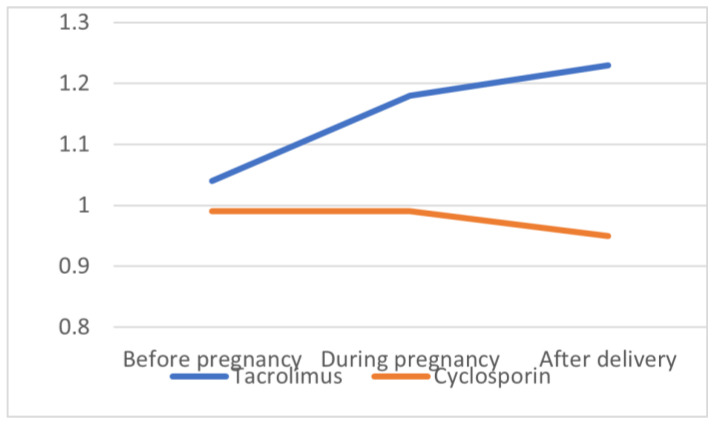
Serum creatinine concentration (mg/dL) before, during, and after pregnancy in patients on tacrolimus-based and cyclosporine-A-based immunosuppressive regimens.

**Table 1 ijerph-19-16352-t001:** Characteristics of study and control group.

Variable	Study Group (*n* = 27)	Control Group (*n* = 110)	*p*
Maternal characteristics:			
Age (years)	35.0 ± 3.5	34.1 ± 4.3	0.31
BMI (kg/m^2^)	24.7 ± 4.2	23.9 ± 7.6	0.19
Nulliparous (%)	13 (48.1)	68 (61.8)	<0.0001
Hypertension before pregnancy (%)	21 (77.8)	0	<0.001
Pregnancy complications:			
Pregnancy-induced hypertension (%)	6 (22.2)	9 (8.2)	0.047
Gestational diabetes (%)	5 (18.5)	7 (6.4)	0.06
Anemia (%)	11 (40.7)	11 (10.0)	<0.0001
Proteinuria (%)	12 (44.4)	3 (2.7)	<0.0001
Intrahepatic cholestasis (%)	0	2 (1.8)	0.98
HCV infection (%)	1 (3.7)	0	0.197
HBV infection (%)	1 (3.7)	1 (0.9)	0.356
Perinatal outcomes:			
Gestational age at delivery (weeks)	34.8 ± 4.2	39.1 ± 1.6	<0.0001
Cesarean section (%)	21 (77.8)	37 (33.6)	<0.0001
Birth weight (g)	2350 ± 878.36	3404 ± 488.37	0.001
Birth weight centiles	38.63 ± 23.47	45.97 ± 27.76	0.171
SGA (<10 percentile) (%)	4 (14.8)	8 (7.3)	0.252

**Table 2 ijerph-19-16352-t002:** Laboratory evaluation of the study group throughout pregnancy.

Variables (Mean ± SD)	Study Group (*n* = 27)
Prepregnancy laboratory values:	
SCr (mg/dL)	1.03 ± 0.24
eGFR (mL/min/1.73 m^2^)	65.5 ± 23.8
Pregnancy (I trimester) laboratory values	
SCr (mg/dL)	1.14 ± 0.38
ALT (U/L)	18.9 ± 0.11
AST (U/L)	17.04 ± 0.83
Hgb (g/dL)	10.84 ± 1.27
Urea (mg/dL)	7.24 ± 1.91
Postpartum Scr (mg/dL)	1.17 ± 0.40

SCr: serum creatinine; eGFR: estimated glomerular filtration rate; ALT alanine transaminase; AST aspartate transaminase, Hgb: Hemoglobin.

**Table 3 ijerph-19-16352-t003:** The comparison of first-trimester screening parameters in study and control groups.

Variable	Study Group (*n* = 27)	Control Group (*n* = 110)	*p*-Value
free β-hCG (mIU/L)	123.06 ± 76.54	50.76 ± 32.71	0.047
free β-hCG MoM	3.47 ± 2.08	1.38 ± 0.85	0.035
PAPP-A (mIU/L)	5.25 ± 4.20	2.48 ± 1.16	0.016
PAPP-A MoM	1.46 ± 0.81	0.98 ± 0.57	0.007
NT (mm)	1.77 ± 0.43	1.70 ± 0.47	0.687
UtA PI	1.06 ± 0.23	1.28 ± 0.53	0.052

**Table 4 ijerph-19-16352-t004:** Comparison of screening parameters and perinatal outcomes between two immunosuppressive therapy regimens.

Variables	Tacrolimus (*n* = 19)	Cyclosporin (*n* = 7)	*p*-Value
NT (mm)	1.91 ± 0.36	1.96 ± 0.31	0.75
UtA PI	1.13 ± 0.37	1.12 ± 0.29	0.95
free β-hCG (mIU/L)	101.01 ± 68.10	157.32 ± 80.50	0.18
free β-hCG MoM	2.87 ± 1.76	3.40 ± 1.30	0.47
PAPP-A (mIU/L)	4.44 ± 3.25	7.88 ± 6.31	0.16
PAPP-A MoM	1.28 ± 0.64	2.02 ± 1.07	0.12
GA at delivery (weeks)	34.16 ± 4.60	36.00 ± 2.70	0.33
Birthweight (g)	2266.05 ± 937.35	2401.43 ± 654.08	0.73
Birthweight percentile	40.79 ± 23.26	26.71 ± 14.49	0.16
SCr before pregnancy (mg/dL)	1.04 ± 0.28	0.99 ± 0.11	0.70
SCr during pregnancy (md/dL)	1.18 ± 0.39	0.99 ± 0.32	0.29
SCr after delivery (mg/dL)	1.23 ± 0.44	0.95 ± 0.16	0.14

## Data Availability

The data presented in this study are available on request from the corresponding author.

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
