# Peer review of "First-Trimester Biochemical Serum Markers in Female Kidney Transplant Recipients—The Impact of Graft Function"

_ijerph, 2022, doi:10.3390/ijerph192316352_

Round 1

Reviewer 1 Report

This is an interesting and clinically useful study, I would congratulate the authors for their work. I think that the data shown support the conclusion regarding the need to develop a special algorithm to interpret first trimester serum markers levels in kidney graft recipients pregnant patients. This is relevant for clinical practice as well as for clinical research. The collateral data on the type of hypertension / vascular damage in this particular subgroup of pregnant patients seem interesting as well.

The study group is small in absolute numbers, which might have impacted the calculated screen-positive rate; on the other hand, it is a large enough group, given the rarity of the condition. A more compelling (maybe prospective?) study design would be desirable, but I think this study is worth publishing as it is.

Author Response

Dear Reviewer,

Thank you for your kind comments. Please find our response below, and we hope you will find it satisfactory.

Comment 1 :

The study group is small in absolute numbers, which might have impacted the calculated screen-positive rate; on the other hand, it is a large enough group, given the rarity of the condition. A more compelling (maybe prospective?) study design would be desirable, but I think this study is worth publishing as it is.

The research on post-transplant perinatal outcomes often enrolls small numbers of patients because the unique population of post-transplant mothers is relatively tiny. Most data on the management derives from single-center studies, case series reports, and a few national and international registries that do not always collect all the relevant data. We sincerely hope for a prospective study in the first-trimester screening and the NIPT performance in a multicenter setting that would allow enrolling more patients in the study group.

Reviewer 2 Report

Please define the abbreviations in the abstract.

Author Response

Dear Reviewer,

Thank you for your comment. The abbreviations in the abstract have now been defined, and we hope it will add more clarity to the abstract. 

Reviewer 3 Report

A very interesting study. Unfortunately, the topic concerns only a very small group of pregnant women. These are 27 women over 11 years. In order to determine the correct medians, usually around 50 women are needed for each week of pregnancy. Moreover, it is certainly not a single method, it has definitely changed over the course of 11 years.

I would see this study as a fundamental basis for requiring the examination of such pregnant women. New methods of non-invasive prenatal testing would already make it possible to distinguish fetal DNA..Alternatively a directly invasive diagnostics. can be recommended.

Is screening for birth defects really carried out in Poland? Abortion is not allowed there, is it?

Author Response

Dear Reviewer,

Thank you for your kind review. Please find below the responses to your comments. 

Comment 1: Unfortunately, the topic concerns only a very small group of pregnant women. These are 27 women over 11 years. In order to determine the correct medians, usually around 50 women are needed for each week of pregnancy.

Response 1: The post-transplant patients are a unique group, and we reviewed all the records on pregnancy and delivery in post-transplant females in our department. We identified 27 kidney recipients with pregnancies that ended in delivery, had complete data on perinatal outcome, underwent the first trimester combined screening, and were assessed as eligible for further analysis. We definitely hope to enroll more numerous groups in the future.

Comment 2:  Moreover, it is certainly not a single method, it has definitely changed over the course of 11 years.

Response 2: All the patients underwent the same procedure: the first-trimester screening was performed according to the Fetal Medicine Foundation algorithm by a sonographer who held the certificate of competence issued by the FMF, and biochemistry results were obtained using the FMF-approved Roche equipment.

Comment 3: I would see this study as a fundamental basis for requiring the examination of such pregnant women. New methods of non-invasive prenatal testing would already make it possible to distinguish fetal DNA..Alternatively, a directly invasive diagnostics can be recommended.

Response 3: We hope for a multicenter study that will allow for the enrollment of more numerous groups of patients. The performance of non-invasive prenatal testing in graft recipients would, of course, require further studies and ideally would also require a multicenter approach.

Comment 4: Is screening for birth defects really carried out in Poland? Abortion is not allowed there, is it?

Response 4: Prenatal diagnostics in Poland is carried out per national guidelines that align with international standards. The legal situation has changed, and the termination of pregnancy is not allowed for fetal indications since 2020. However, this does not mean that pregnant patients are declined knowledge of the diagnosis of fetal anomalies in pregnancy. Invasive procedures are also offered when indicated.

We sincerely hope you will find our answers satisfactory.